# WASH: Train your Ensemble with Communication-Efficient Weight Shuffling, then Average

## Abstract

The performance of deep neural networks is enhanced by ensemble methods, which average the output of several models. However, this comes at an increased cost at inference. Weight averaging methods aim at balancing the generalization of ensembling and the inference speed of a single model by averaging the parameters of an ensemble of models. Yet, naive averaging results in poor performance as models converge to different loss basins, and aligning the models to improve the performance of the average is challenging. Alternatively, inspired by distributed training, methods like DART and PAPA have been proposed to train several models in parallel such that they will end up in the same basin, resulting in good averaging accuracy. However, these methods either compromise ensembling accuracy or demand significant communication between models during training. In this paper, we introduce WASH, a novel distributed method for training model ensembles for weight averaging that achieves state-of-the-art image classification accuracy. WASH maintains models within the same basin by randomly shuffling a small percentage of weights during training, resulting in diverse models and lower communication costs compared to standard parameter averaging methods.

## 1 Introduction

In order to enhance the accuracy of a given class of models, the answers of multiple instances trained in parallel can be aggregated via model *ensembling*. This can lead to significant improvements in modern deep learning models (12), increasing the generalization ability. However, this comes at the cost of evaluating multiple instances of a given model during inference. This increases both memory and computational requirements, resources that can be critical for on-device inference (32). To solve this problem, the population of models can be fused into a single model to obtain both the generalization improvements of ensembling and the inference cost of a single model. Since independent models can be linearly connectable (14), a simple technique is to average the weights of the different models to obtain a fused model (52).

However, there are limits to this method. For models that are too dissimilar, the performance of the averaged model may be no better than chance (19). To mitigate this, the ensemble can either use a pre-trained network as a starting point (34) or ensure that models share part of their optimization path (14). However, reducing ensemble diversity too much comes at the expense of performance (see Figure 6 of (12)), revealing a trade-off between model diversity and weight averagability. Inspired by distributed training, techniques such as DART (20) and PAPA (21) have been proposed to train a population of models in parallel on heterogeneous data while communicating to balance this trade-off. DART, similar to LocalSGD (44), periodically averages all models to avoid model divergence. PAPA controls the diversity of the models more finely by pushing them toward the averaged parameters using an Exponential Moving Average (EMA) like EASGD (55), achieving better performance. In

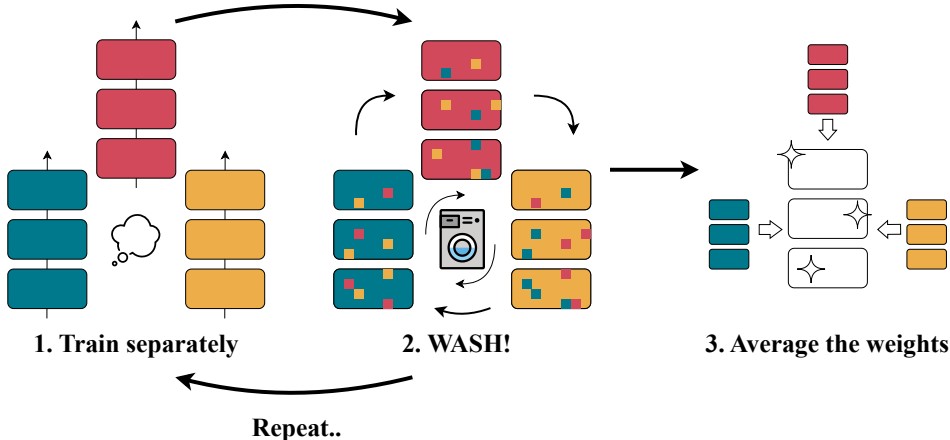

**1. Train separately**  **2. WASH!**  **3. Average the weights**

**Repeat..**

Figure 1: **Representation of training with WASH**. A population of models is being trained separately. **(1)** After each training step, **(2)** a small percentage of the parameters are permuted between models. **(3)** At the end of the training, the model weights are averaged, resulting in a high performance model.

particular, they show that training a population in this way results in models that generalize better than a single model trained with the same compute as the entire population, demonstrating the potential of these distributed approaches. However, existing methods require a regular computation of the average model using an all-reduce operation, either to periodically remove any diversity in the population (20) or, in the case of PAPA, to compute an EMA of the average. This results in a high communication cost during the parallel training of the model population (36), which hampers the scalability of these approaches as the population size increases (35).

In this work, we propose a novel distributed method to train a population of models in parallel while keeping their weights within the same basin. It requires a fraction of the communication cost of PAPA but exhibits greater model diversity during training, increasing the final averaging accuracy. Our main idea is to shuffle parameters between models during training, forcing them to learn using the others' parameters. We refer to this idea as "parameter shuffling". A permutation is chosen randomly, and the models will communicate their parameters peer-to-peer according to the permutation. The use of a permutation is distinct from the notion of weight permutation of (1), which is within one model. We denote our method, which achieves **W**eight **A**veraging using parameter **SH**uffling, as **WASH**, and represent it schematically in Fig. 1.

**Contributions.** Our work makes the following contributions: **(1)** We propose a novel method for the training of a population of models that can be weight-averaged, which we call WASH (**W**eight **A**veraging using parameter **SH**uffling). By shuffling a small number of parameters between models during training, the resulting population can be weight-averaged into a high-performance model for a fraction of the communication volume of methods such as PAPA. **(2)** We find that WASH provides state-of-the-art results on image classification tasks, resulting in models with performance at the level of ensembling methods, while requiring only a single network at inference time. **(3)** We provide experiments to better understand the improvement provided by WASH, in particular how WASH implicitly reduces the distance between models in the population while preserving diversity. **(4)** We perform different ablations of our method to show the impact of shuffling. **(5)** At the time of publication, we will release an implementation of WASH on an open repository.

## 2 Related work

**Ensemble and weight averaging.** By combining predictions from multiple models, ensemble methods significantly improve the ability of a predictive system to make accurate generalizations (8; 26), while reducing the variance of the estimator (4).

This variance reduction is particularly effective when errors are uncorrelated and models exhibit diversity, that is, they do not fail simultaneously on the same instances (17; 12). However, ensembles require additional passes through each model for inference, leading to increased computational costs.

This cost can become prohibitive for large numbers of models. As a remedy, under certain conditions, models can be averaged together to remove the computational burden during inference. Averaging the weights of models was first explored in simple linear (27) and convex scenarios (38; 3). In deep learning, (19) establishes that weight averaging is a first-order approximation of the ensemble when models are close in weight space. Notably, simple averaging of multiple points along the SGD trajectory leads to better generalization. Following mode connectivity (16; 14) and the observation that many optima of independent models are connectable, (2; 51) propose learning simplexes in the parameter space with a regularisation penalty to encourage diversity in the weight space, and (53; 39) propose to train several model branches with different last-layer initialization and hyperparameters simultaneously. These models are later averaged to improve generalization and reduce inference costs. However, for these models to be amenable to weight averaging, they generally must start with the same pre-trained initialization (34), which can reduce the diversity between models. To alleviate this problem, neuron alignment techniques (42; 1; 37; 18) match the units of multiple networks to make them amenable to weight averaging, but they rarely work in practical scenarios (22) and often achieve performance below that of the individual models. DART (20) and Branch-Train-Merge (BTM) (28) propose a three-phase training pipeline. The process begins with an initial shared training phase, followed by the parallel training of multiple models, each diversified by different data domains or different data augmentations. Finally, these models are merged into a single model. They find that iterative refinement of the last 2 stages improves the overall optimization trajectory and improves generalization. To enhance the diversity among the models, PAPA (21) proposes to gradually adjust the model weights towards the population average throughout the training process, starting from random initialization. However, these approaches can result in significant communication costs during training. Conversely, WASH addresses high communication costs by permuting only a small fraction of parameters between models during training, while ensuring that branches remain accessible for weight averaging at the end.

**Distributed and federated learning.** In the distributed training of deep learning models, the tradeoff between communication and model performance is a core concern (35; 23), and finding methods to efficiently mitigate some of the communication costs is a recurring theme in different research areas (46; 13). For example, since communication overhead is a key concern in decentralized optimization, it has been shown in this literature that for training models in a data-parallel setting with a limited communication budget, a key metric to observe is the average distance to consensus (24; 40; 45; 49; 33). The techniques discussed earlier for training a population of models for weight averaging are similar to methods in the LocalSGD (44; 30) and Federated Learning (31; 23; 29) literature. The training in DART and BTM is similar to LocalSGD training, where models are periodically averaged after several computational steps. PAPA, which uses an EMA of the averaged model to gradually move the models towards consensus, is similar to methods such as EASGD (55) or SlowMo (48). Just averaging a population at the end of training, as in BTM, has also been proposed for LocalSGD (43), and cross-gradient aggregation (11) can be seen as a way of locally shuffling gradients. Federated learning also uses techniques discussed previously for model merging (47; 54; 6). Finally, our method can be thought of as training a global model, where each local model randomly chooses from a subset of parameters when shuffling. This can be linked to Bayesian learning (15), especially for federated learning (50; 9), or federated subnetwork training (10; 41).

## 3 Parameter shuffling in an ensemble for weight averaging

**Motivation of our training procedure.** We aim to balance the benefits of model ensembling with the computational efficiency of using a single model for inference via weight averaging. In other words, our objective is to produce a single model resulting from the ensembling. A set of $N$ model parameters $\{\theta_n\}_{n \leq N} \subset \mathbb{R}^d$ are trained in parallel on the same dataset, with different data ordering and possibly different data augmentations and regularizations. To avoid divergence between the models, PAPA applies an EMA every $T$ training steps and produces the following update

$$\tilde{\theta}_n \leftarrow \alpha \theta_n + (1 - \alpha)\bar{\theta}, \tag{1}$$

where $\bar{\theta} \triangleq \frac{1}{N} \sum_{n=1}^{N} \theta_n$ represents the average of the model weights, also called the *consensus*, and $\alpha \in [0, 1]$ is weighted according to the learning rate. Despite its advantages, this method has drawbacks, including the need for synchronized global communication across all models, which can

**Algorithm 1** Training with WASH

1: **Input:** Datasets $D_i$, number of models $N$, initial parameters $\theta_0$, training steps $T$, number of layers $L$, base probability $p$
2: Initialize parameters $(\theta_n)_n \leftarrow \theta_0$ and optimizers $\text{OPT}_i$
3: **for** $t = 1$ to $T$ **do**
4:     *# Training step*
5:     **for** $n = 1$ to $N$, in parallel **do**
6:        $(x_n, y_n) \leftarrow D_n$          *# Sample data*
7:        $\theta_n \leftarrow \text{OPT}_n(x_n, y_n, \theta_n)$    *# Update the model $n$*
8:     *# Shuffling step*
9:     **for** layer $l = 0$ to $L - 1$ **do**
10:        **for** parameter $\theta^i$ in layer $l$ **do**
11:           **With** probability $p(1 - \frac{l}{L-1})$,
12:             $\pi_i \leftarrow$ Random permutation
13:             $(\theta_n^i)_n \leftarrow (\theta_{\pi_i(n)}^i)_n$   *# Send and permute the parameter*
14: **Output:** the averaged model $\frac{1}{N} \sum_{n=1}^{N} \theta_n$

be inefficient, and the potential reduction in model diversity due to the consensus constraint, which may reduce model expressiveness. Indeed, we observe that after each update

$$\sum_n \|\tilde{\theta}_n - \bar{\theta}\|^2 = \alpha^2 \sum_n \|\theta_n - \bar{\theta}\|^2 < \sum_n \|\theta_n - \bar{\theta}\|^2, \tag{2}$$

which shows that the EMA step of methods such as PAPA directly reduces the distance of the models from the consensus and hinders their diversity.

**Proposed method: WASH.** To address these challenges, we propose the following stochastic parameter shuffling step instead of the EMA, defined for each individual parameter $\hat{\theta}_n^j \in \mathbb{R}$ of a model $\theta_n = [\theta_n^j]_{j=1}^d$ by

$$\hat{\theta}_n^i \leftarrow \begin{cases} \theta_{\pi_i(n)}^i & \text{with probability } p, \\ \theta_n^i & \text{otherwise,} \end{cases} \tag{3}$$

where $\pi_i$ denotes a random permutation of the indices $\{1, ..., N\}$, chosen uniformly at each iteration for each parameter index $i \in \{1, ..., d\}$, and independently from the Bernoulli variable of Eq. (3). Notably, this parameter shuffling reduces in expectation to

$$\mathbb{E}[\hat{\theta}_n] = (1 - p)\theta_n + p\bar{\theta}. \tag{4}$$

Thus, WASH aligns, in expectation, with the EMA of Eq. (1) for $p = (1 - \alpha)$. The expected number of parameters communicated by each model at each step is thus $p \times d$ while for PAPA, each model communicating all of its parameters every $T$ steps, this amounts to $\frac{d}{T}$. Thus, $p \ll \frac{1}{T}$ results in a significantly reduced communication overhead favorable to WASH. However, the model diversity is higher, because WASH preserves the consensus distance, as shown by

$$\sum_n \|\hat{\theta}_n - \bar{\theta}\|^2 = \sum_n \sum_i (\hat{\theta}_n^i - \bar{\theta}^i)^2 = \sum_i \sum_n (\theta_n^i - \bar{\theta}^i)^2 = \sum_n \|\theta_n - \bar{\theta}\|^2. \tag{5}$$

Still, note that the following optimization step on the shuffled parameters will affect the consensus distance, as we will see later.

**Layer-wise adaptation via WASH.** Recognizing that different network layers may require different levels of adaptation due to their roles and dynamics, we introduce a layer-specific probability adaptation. Assuming $L$ layers in the network, for each layer $l$ (where $0 \le l < L$) we set

$$p_l = p\left(1 - \frac{l}{L-1}\right), \tag{6}$$

Table 1: **Communication volume and inference costs** of four training techniques. The baseline Ensemble is trained separately, but requires a linearly increasing inference cost. In our experiments, we set the base probability of WASH and WASH+Opt to 0.001 and 0.05, respectively, when training on CIFAR-10/100 or ImageNet, resulting in a reduction in communication volume compared to PAPA.

| | Communication volume | | |
|---|---|---|---|
| **Technique** | **CIFAR-10/100** | **ImageNet** | **Inference cost** |
| Ensemble | 0 | 0 | $N$ |
| PAPA | 1 | 1 | 1 |
| WASH | $^1/_{200}$ | $^1/_4$ | 1 |
| WASH+Opt | $^1/_{100}$ | $^1/_2$ | 1 |

where $p$ is a base probability. In other words, the parameters of the first layer have a shuffling probability of $p$, while the parameters of the last layer are never shuffled. This adaptation ensures that deeper layers, which are typically slower to train and more sensitive to the input features, undergo fewer permutations than the more generalizable early layers. This strategy not only preserves the specificity required by the early layers, but also cuts the overall communication overhead in half.

**Full procedure.** Alg. 1 presents the training of a population of $N$ models using WASH. Starting from the same initialization, our training procedure alternates between local gradient computation and shuffling communication. At inference, we simply average the weights of the models to obtain a single model with parameters $\bar{\theta}$. Note that techniques such as REPAIR (22) or activation alignment (1) could be incorporated to improve the alignment of the models, but we found them to be unnecessary to achieve high accuracy and kept our evaluation framework minimal for the sake of simplicity.

# 4 Experiments

**Training methods.** We present the capabilities of WASH for training a population of neural networks on standard image classification tasks. As a Baseline, we consider a population trained separately, with each model working on a different dataset order and different data augmentations and regularization (if they are used). This is the same baseline as (21), only starting from the same initialization, but we found that this change had no significant impact on performance. We also compare WASH to PAPA (21) on the same tasks (with PAPA however using models with a different initialization), to show our improvement despite requiring a fraction of the communication cost. We do not provide comparisons with DART (20) or the variants of PAPA as their performances are generally inferior (21). We also propose a variant of WASH called WASH+Opt, which also permutes the optimizer state associated with the shuffled parameter (in our case, the momentum of SGD), doubling the communication volume. For simplicity, we do not permute or recompute the running statistics of the BatchNorm layers.

**Communication cost.** Training with PAPA requires computing an all-reduce operation on all of the model parameters every $T = 10$ training steps. In comparison, WASH requires, in expectation, a shuffling of $p/2$ of the model parameters at each training step. Thus, by keeping a base probability $p \leq 0.2$, WASH results in a more communication-efficient training. In practice, in our experiments, $p$ will be 0.001 or 0.05, ensuring a reduction in communication volume of 200 or 4.

**Evaluation strategy.** After training, the resulting population of models obtained can be evaluated in three different ways. As a baseline, the performance of the population can be evaluated as an Ensemble, averaging the predictions of the models. The parameters of the models can be averaged to obtain a single model, which we refer to as Averaged. This is equivalent to UniformSoup in (52) or AvgSoup in (21) for example. More elaborate averaging methods have been proposed, such as GreedySoup (52), which averages an increasing number of models (in order of validation accuracy) until averaging no longer improves accuracy. We report the accuracy of the Ensemble and Averaged model for all training techniques, as well as the GreedySoup accuracy of the Baseline. As in (21), we find that the GreedySoup accuracy corresponds to the accuracy of a single model for the Baseline and that the Averaged model accuracy outperforms the GreedySoup model for the other techniques, and

Table 2: **Ensemble and Averaged Model accuracy for a heterogeneous population of models; trained with varying data augmentations and regularizations.** We compare models trained separately (Baseline), with PAPA, or with our method WASH and its variant WASH+Opt. We also report the GreedySoup accuracy for the Baseline models. The best Ensemble (black) and Averaged (blue) accuracy are reported in bold. Except on CIFAR-10, WASH and in particular WASH+Opt provide the best performance for the final Averaged Model, with performances comparable to the Ensemble of models for a fraction of the inference cost

| Method | | Baseline (trained separately) | | | PAPA | | WASH (ours) | | WASH+Opt (ours) | |
|---|---|---|---|---|---|---|---|---|---|---|
| Config | #N | Ensemble | Averaged | GreedySoup | Ensemble | Averaged | Ensemble | Averaged | Ensemble | Averaged |
| **CIFAR-10** | | | | | | | | | | |
| VGG-16 | 3 | 95.98±.42 | 10.00±.00 | 95.26±.05 | **96.12±.34** | **96.13±.24** | 95.89±.23 | 95.97±.24 | 95.91±.36 | 95.85±.27 |
| | 5 | **96.28±.40** | 10.00±.00 | 95.42±.10 | 96.24±.17 | **96.21±.13** | 96.15±.10 | **96.20±.10** | 96.00±.21 | 96.04±.14 |
| | 10 | **96.47±.07** | 10.00±.00 | 95.39±.24 | 96.32±.13 | **96.31±.13** | 96.27±.10 | 96.18±.13 | 96.14±.08 | 96.20±.05 |
| ResNet18 | 3 | 97.15±.28 | 10.17±.29 | 96.62±.38 | **97.33±.05** | **97.24±.05** | 97.21±.19 | 97.19±.17 | 97.22±.07 | **97.25±.14** |
| | 5 | 97.33±.08 | 10.09±.16 | 96.61±.03 | **97.35±.12** | **97.31±.06** | 97.21±.10 | 97.25±.12 | 97.18±.09 | 97.16±.07 |
| | 10 | **97.59±.01** | 9.26±1.28 | 96.79±.14 | 97.39±.13 | **97.34±.06** | 97.30±.10 | 97.28±.04 | 97.20±.13 | 97.16±.13 |
| **CIFAR-100** | | | | | | | | | | |
| VGG-16 | 3 | **80.36±.15** | 1.00±.00 | 77.92±.22 | 78.89±.10 | 78.77±.16 | 79.10±.88 | 79.05±.68 | 79.15±.61 | **79.15±.41** |
| | 5 | **81.32±.56** | 1.00±.00 | 77.81±.25 | 79.51±.38 | 79.24±.43 | 79.65±.27 | 79.39±.21 | 79.75±.21 | **79.71±.20** |
| | 10 | **82.24±.15** | 1.00±.00 | 77.83±.65 | 79.95±.11 | 79.64±.13 | 80.05±.18 | 79.70±.25 | 80.03±.11 | **79.76±.13** |
| ResNet18 | 3 | **82.84±.48** | 1.00±.01 | 80.06±1.5 | 81.58±.12 | 81.53±.13 | 81.91±.34 | 81.90±.36 | 81.99±.06 | **82.08±.09** |
| | 5 | **83.72±.49** | 1.00±.00 | 80.72±.52 | 82.09±.30 | 82.01±.34 | 82.16±.42 | 81.97±.28 | 82.35±.17 | **82.17±.15** |
| | 10 | **84.18±.20** | 1.00±.00 | 80.61±.43 | 82.32±.09 | 82.15±.14 | 82.43±.32 | **82.31±.38** | 82.42±.31 | 82.18±.22 |
| **ImageNet** | | | | | | | | | | |
| ResNet50 | 3 | **76.16±.28** | 0.10±.00 | 74.15±.11 | 75.62±.15 | * | 74.39±.14 | **74.34±.18** | 74.30±.22 | 74.18±.26 |
| | 5 | **76.68±.06** | 0.10±.00 | 74.47±.06 | 75.80±.21 | * | 74.63±.11 | **74.59±.07** | 74.44±.21 | 74.39±.21 |

thus chose not to report it. We summarize in Tab. 1 the communication volume and inference costs required to train a separate Ensemble of models, or to train with PAPA, WASH, or WASH+Opt.

## 4.1 Main experiments

**Experimental setup.** We showcase the performance of WASH for training neural networks on image classification tasks on the CIFAR-10, CIFAR-100 (25), and ImageNet (7) datasets. We use the same training framework as (21) for a fair comparison. We train a population of $N$ models for $N \in \{3, 5, 10\}$, on the ResNet-18, 50 and VGG-16 architectures. 2% of the training data is kept as validation for computing the GreedySoup. As in (21), we consider one framework with heterogeneous models, learning with different data augmentations and regularizations, and one homogeneous setting with no data augmentations except random cropping and flipping, in addition to the different dataset shuffling. Details are presented in the Appendix. The models are trained with SGD with momentum, a weight decay of $10^{-4}$, and a cosine annealing scheduler with initial and minimum learning rates of $0.1$ and $10^{-4}$. For CIFAR-10/100, we train over 300 epochs with a batch size of 64, and 90 epochs with a batch size of 256 for ImageNet. For WASH and WASH-Opt we initialize the models with the same parameters and choose $p$ with cross-validation to be equal to $0.001$ when training on CIFAR-10/100 or $0.05$ for ImageNet. We do not require any alignment technique such as REPAIR (22).

**Main results.** Tab. 2 and Tab. 3 correspond to the heterogeneous and homogeneous settings, respectively. We report the test accuracies as the average of 3 runs for the Ensemble of models, the Averaged model, and the GreedySoup for the Baseline (equivalent to the best model). Consistent with the findings of (21), we find that networks trained separately have a high Ensemble accuracy, but perform as random when averaged. On CIFAR-10/100, methods like PAPA and WASH result in lower Ensemble accuracy but almost no difference between the Ensemble and Averaged accuracies. In general, WASH and WASH+Opt outperform PAPA, even though they require less communication. On ImageNet, our parallelization procedure results in a slightly lower Baseline accuracy and we were not able to reproduce PAPA's baseline, possibly due to a mistake in their reported hyperparameters (See the Appendix for experiments on ImageNet32x32). The WASH Averaged model achieves high accuracy, like previously. Both of our methods reduce the gap with the accuracies of the baseline Ensemble, indicating that WASH hinders less the diversity of the population of models while maintaining weight averagability. However, a gap still remains, which may be inherent to the models

Table 3: **Ensemble and Averaged Model accuracy for a homogeneous population of models.** We compare models trained separately (Baseline), with PAPA, or with our methods WASH and WASH+Opt. The best Ensemble (black) and Averaged (blue) accuracy are reported in bold. We observe the same results in this setting, with WASH in particular coming close to the Ensemble performance. We report the accuracy for models trained with PAPA on ImageNet with $T = 1$.

| Method | | Baseline (trained separately) | | | PAPA | | WASH (ours) | | WASH+Opt (ours) | |
|---|---|---|---|---|---|---|---|---|---|---|
| Config | #N | Ensemble | Averaged | GreedySoup | Ensemble | Averaged | Ensemble | Averaged | Ensemble | Averaged |
| **CIFAR-10** | | | | | | | | | | |
| **VGG-16** | 3 | **94.93±.06** | 10.00±.00 | 93.60±.41 | 94.38±.14 | 94.34±.18 | 94.41±.23 | **94.58±.17** | 94.45±.05 | 94.47±.02 |
| | 5 | **95.29±.05** | 10.00±.00 | 93.82±.30 | 94.55±.12 | 94.58±.12 | 94.72±.08 | **94.70±.17** | 94.63±.11 | **94.68±.14** |
| | 10 | **95.23±.06** | 10.00±.00 | 93.82±.06 | 94.79±.18 | **94.78±.20** | 94.66±.03 | 94.54±.07 | 94.71±.07 | 94.61±.13 |
| **ResNet18** | 3 | **96.14±.10** | 10.00±.00 | 95.42±.27 | 95.89±.04 | **95.89±.06** | 95.77±.12 | 95.77±.17 | 95.85±.04 | **95.87±.10** |
| | 5 | **96.19±.16** | 10.00±.00 | 95.31±.09 | 95.99±.08 | **95.99±.08** | 95.96±.08 | **95.98±.05** | 95.94±.12 | **95.98±.12** |
| | 10 | **96.34±.02** | 10.00±.00 | 95.26±.11 | 96.10±.25 | **96.11±.24** | 96.08±.07 | **96.12±.09** | 96.07±.07 | 96.08±.14 |
| **CIFAR-100** | | | | | | | | | | |
| **VGG-16** | 3 | **77.63±.24** | 1.00±.00 | 73.76±.35 | 75.10±.11 | 75.09±.16 | 76.30±.37 | **76.04±.58** | 76.04±.03 | 75.96±.18 |
| | 5 | **78.52±.10** | 1.00±.00 | 73.76±.18 | 75.56±.16 | 75.55±.14 | 76.63±.27 | **76.48±.23** | 76.64±.15 | 76.13±.18 |
| | 10 | **79.26±.06** | 1.00±.00 | 73.99±.26 | 76.24±.44 | 76.26±.43 | 77.06±.12 | **76.43±.18** | 76.72±.15 | 75.94±.26 |
| **ResNet18** | 3 | **79.54±.17** | 1.00±.00 | 76.84±.54 | 77.83±.26 | 77.86±.30 | 78.90±.17 | **78.76±.25** | 78.66±.08 | 78.56±.21 |
| | 5 | **80.11±.23** | 1.00±.00 | 76.83±.45 | 77.94±.16 | 77.92±.19 | 79.24±.32 | 79.09±.43 | 79.32±.19 | **79.19±.15** |
| | 10 | **80.55±.13** | 1.00±.00 | 76.80±.41 | 78.40±.15 | 78.44±.22 | 79.65±.17 | **79.43±.16** | 79.34±.34 | 79.19±.45 |
| **ImageNet** | | | | | | | | | | |
| **ResNet50** | 3 | **75.7 ± .15** | 0.10±.00 | 73.2 ± .15 | 73.4 ± .30 | 73.4 ± .29 | 74.0 ± .12 | **73.8 ± .05** | 73.9 ± .15 | 73.8 ± .11 |

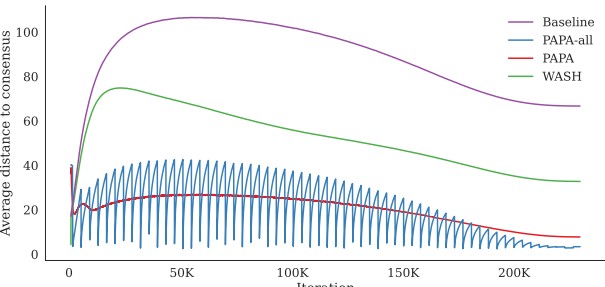

Figure 2: **Average distance to the consensus (i.e. the averaged model)** during training for a heterogeneous population of 5 models trained on CIFAR-100, either separately, with PAPA, PAPA-all, or our method WASH. Starting from consensus, the models initially diverge from each other before converging back again during convergence, mainly due to weight decay. Models trained with WASH have a smaller distance to consensus than those trained separately, allowing them to be averaged without loss of performance. By training with PAPA-all (i.e., averaging to a single model every few epochs), the models are not able to reach the same diversity as WASH between these averaging steps. Finally, the EMA of PAPA has a strong pulling effect toward consensus, resulting in a distance similar to that of PAPA-all. The wiggle in the curve is due to the immediate reduction in distance caused by the EMA steps

being in the same basin. WASH and WASH+Opt have very similar results, with the simpler WASH being better in the homogeneous case and WASH+Opt being better in the heterogeneous case.

## 4.2 Why do shuffling parameters help?

In this section, we propose to explain the improvement provided by our parameter shuffling over previous mechanisms such as BTM, DART or PAPA, which focus on parameter averaging. First, we show that models trained with WASH have a smaller distance to consensus than models trained separately. We then argue that, despite this, WASH is a weak perturbation on the training of the models and that it induces diversity in the models.

**Reducing distance to consensus.** To better analyse the diversity of the models trained with WASH, we propose to report the distance of the models to the consensus (the averaged model) during training, as a proxy for the diversity metric. (19; 53) showed that the difference between the Ensemble and

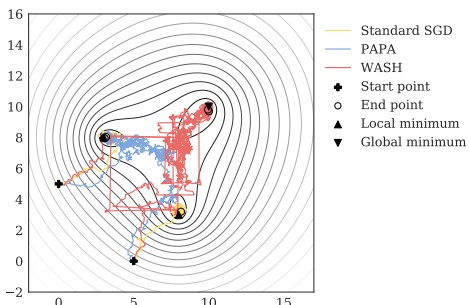

Figure 3: **2D optimization example.** We train 2 points with SGD on a simple loss function with 2 local and 1 global minima (up and down triangles). The two models are trained from two different starting points (plus signs). When the points are trained separately (yellow), they converge to their closest local minimum (yellow circles). When trained with PAPA (blue), the points reach a consensus but then converge to one of the local minima (blue circles). When trained with WASH (red), the shuffling (seen by the horizontal and vertical lines in the trajectory) allows for more diversity in the optimization path, and the points both reach the global minimum (red circles).

the Averaged models depends on the distance between the models. We present in Fig. 2 the average distance of the models to the consensus, for models trained separately, with PAPA, PAPA-all, or with WASH. PAPA-all is a variant of PAPA that is functionally identical to DART. The idea is to average the weights every few epochs before allowing the models to diversify again. We observe that WASH results in a consistently lower distance to consensus than the baseline, even though it explicitly leaves the distance to consensus unchanged during the shuffling step, and only shuffles a small number of parameters. Thus, the smaller distance at the end of the training explains why the averaging of the parameters does not lead to a decrease in performance. In comparison, PAPA-all (i.e. DART) results in alternating phases where the models diversify before being averaged, and we observe that the models are not able to reach the diversity of WASH. Similarly, the EMA of PAPA has a strong pulling effect and results in an average diversity similar to that of PAPA-all. Thus, we find that models trained with WASH have a higher diversity than models trained with PAPA or PAPA-all, while being close enough that averaging them does not cause a loss in performance. More generally, we show in Fig. 6 of the Appendix that different interpolations of models trained with WASH result in a similar performance, demonstrating that they all lie in the same loss basin.

**Encouraging diversity.** WASH can be considered as a weak perturbation of the models: parameter shuffling affects the models less than parameter averaging or the EMA of PAPA, since only a few parameters are affected at a time and the consensus distance is unaffected. Furthermore, parameter shuffling increases the diversity of trajectories seen by the models. We illustrate this with a toy example where two points are jointly trained with SGD on a 2D loss function with 2 local minima and 1 global minimum, either separately, with PAPA, or with WASH. The trajectories corresponding to each method are shown in Fig. 3. Training the two points separately causes them to converge to a separate local minimum (i.e. a different basin). Training with PAPA allows the two points to reach a consensus, but they converge together to a local minimum. In contrast, by training with WASH, we show that both points reach the global minimum, as the shuffling allows for a greater diversity of points to optimize with. We provide more details in the Appendix.

### 4.3 Ablations

In this section, we present ablations to better understand the effect of the parameter shuffling, varying the layer-wise probability adaptation, the base probability value, and the shuffling period. In all cases, we consider 5 ResNet-18 models trained on CIFAR-100 in a heterogeneous environment.

**Layer-wise adaptation variations.** For WASH, we found that decreasing probability with depth gave the best results. We show in Tab. 4 of the Appendix the performances for alternatives where the probability either remains constant or increases with depth. We find lower performances for both alternatives. In Fig. 4 we show the distances of the models to the consensus for all three schedules. More specifically, we report the distances for different slices of the models' parameters, showing

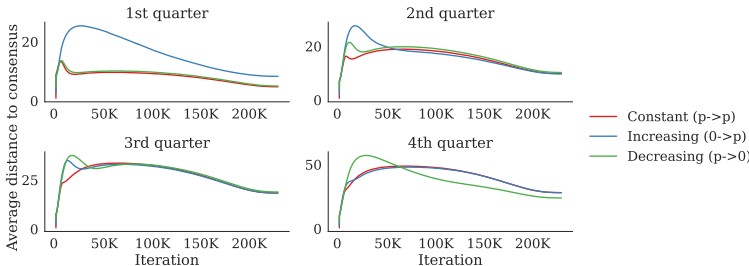

Figure 4: **Average distance to the consensus for different layer-wise adaptations of WASH**, for different slices of the model parameters. Keeping the probability constant across layers ensures the lowest distance to consensus for the first quarters. Surprisingly, in the last quarter of parameters, the 'decreasing probability' adaptation, despite starting with a higher distance to consensus, shows a lower distance to consensus later in training; even though shuffling is less frequent than in the other schedules. The 'increasing probability' adaptation shows how early layers are useful for shuffling.

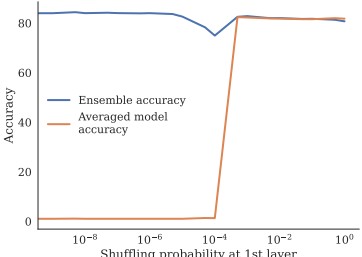

(a) **Ensemble and Averaged accuracy for varying base probability values.** We observe a phase transition as the base probability increases between a phase where permuting does not improve the averaged model accuracy and a phase where the ensemble accuracy is equal to the averaged model accuracy. Between the phases, the ensemble accuracy decreases.

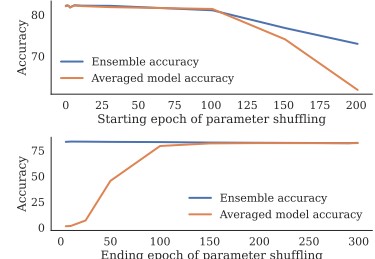

(b) **Ensemble and Averaged accuracy depending on the starting or ending epoch of the shuffling.** The parameter shuffling is beneficial both at the beginning and at the end of training. Note that ending early, at epoch 150 out of 300, has less impact on performance than starting permuting at epoch 150, showing that WASH is more important early in training.

Figure 5: **Ablations of WASH**

the effect of shuffling as a function of depth. As predicted, shuffling all layers equally results in the lowest distance to the consensus, except for the last quarter of parameters. Here, surprisingly, our base 'decreasing' adaptation shows a lower distance to the consensus despite less frequent shuffling. We also observe a particularly strong effect of the shuffling for the early layers, as the distance in the first quarter is more pronounced between the 'increasing' curve and the others.

**Base probability variation.** We present in Fig. 5a the Ensemble and Averaged for different values of $p$, the base shuffling probability of the first layer. Rather than a smooth increase in the accuracy of the Averaged model, we observe a phase transition between a phase where the accuracy of the Averaged model is not improved by the shuffling and a sudden increase in the accuracy where it reaches the accuracy of the Ensemble. Just before the transition, the accuracy of the Ensemble decreases, before increasing again back to its previous performance. The accuracy decreases only slightly even when the shuffling probability is increased to 1, indicating the resilience of the models to heavy shuffling.

**Shuffling is beneficial at every step.** Finally, we propose to show the impact of the parameter shuffling at different steps of the training by varying the epoch at which the shuffling either starts or stops. In Fig. 5b, we show that there is no improvement by having a warmup or slowdown period in parameter shuffling, indicating that all phases of the training are improved by WASH. Furthermore, stopping parameter shuffling early results in a much smaller loss of Averaged accuracy compared to starting shuffling late. In other words, shuffling at the beginning of training before the models start to converge is more impactful as the models may still reside in different loss basins.

# 5    Conclusion

We proposed a novel distributed training method, WASH, which aims to train a population of models in parallel. These models are averaged at the end of training to obtain a high performance model with accuracies close to the ensemble accuracy for a fraction of the inference cost. Our method requires a fraction of the communication cost of similarly performing techniques, while achieving state-of-the-art results for our weight-averaged models. We show that our novel parameter shuffling does not explicitly reduce the distance between models while increasing the diversity of optimization paths seen by the population. Nevertheless, we find that the distance between our models is smaller than if they were trained separately, allowing them to be averaged at the end of training.

## Acknowledgements

This work was supported by Project ANR-21-CE23-0030 ADONIS, EMERG-ADONIS from Alliance SU, and Sorbonne Center for Artificial Intelligence (SCAI) of Sorbonne University (IDEX SUPER 11-IDEX-0004). This work was granted access to the AI resources of IDRIS under the allocations 2023-A0151014526 made by GENCI.

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

# 6 Appendix

**2D optimization example** The loss function we consider is a heavily simplified version of the Ackley function. With a minima in $(x_m, y_m)$ defined by

$$g(x, y, x_m, y_m, \lambda) = \exp\left(-\lambda\sqrt{0.5((x - x_m)^2 + (y - y_m)^2)}\right), \tag{7}$$

the function we consider in our example is

$$f(x, y) = -10g(x, y, 10, 10, 0.1) - 5g(x, y, 8, 3, 0.3) - 5g(x, y, 3, 8, 0.3). \tag{8}$$

This function has a 2 local minima in $(3, 8)$ and $(8, 3)$ and a global minimum in $(10, 10)$. In all three cases, the starting points are $(0, 5)$ and $(5, 0)$. We compute SGD by first computing the exact gradient of the function and then adding Gaussian noise to the gradient. The learning rate is $0.1$ and we optimize for $1000$ steps. For PAPA, we consider $\alpha = 0.99$. For WASH, the shuffling probability is equal for both coordinates and equal to $0.01$.

**Interpolation heatmap** Here, we propose to display a heatmap showing the accuracy of more varied interpolations between 5 models trained separately, with WASH, or WASH+Opt. We observe how WASH and WASH+Opt trained models converge to the same loss bassin, and that a large number of possible interpolations result in a high accuracy. The heatmaps are presented in Fig. 6. The performance of each individual model is represented at the five extremities of the heatmaps (see a. notably). Then, each other performance represented in the heatmap circle is for a model with its parameters interpolated between the 5 models. The interpolation weights are computed by normalizing the distance (from a Gaussian kernel) between the point in the circle and the 5 points at the extremities. The center of the heatmap represents an equally weighted average of the models, as implemented in WASH and the other methods considered.

**Layer-wise adaptation variants performance** We showcase in Tab.4 the performance of the three variants of layer-wise adaptations of WASH.

**Augmentations and regularization used** We follow the same data augmentations and regularizations used in (21) for a fair comparison. We use Mixup (random draw from {0, 0.5, 1.0} for CIFAR-10/100 or from {0, 0.2} for ImageNet), Label smoothing (random draw from {0, 0.05, 0.1} for CIFAR-10/100 or from {0, 0.1} for ImageNet), CutMix (random draw from {0, 0.5, 1.0} for CIFAR-10/100 or from {0, 1.0} for ImageNet) and Random Erasing (random draw from {0, 0.15, 0.35} for CIFAR-10/100 or from {0, 0.35} for ImageNet).

For our experiments, we required a single A100 GPU for up to 14 hours to train up to a population of 10 models, and up to 40 hours for a population of 20 models. Similarly, we required 16 A100 GPUs to train in parallel a population of 5 models on ImageNet.

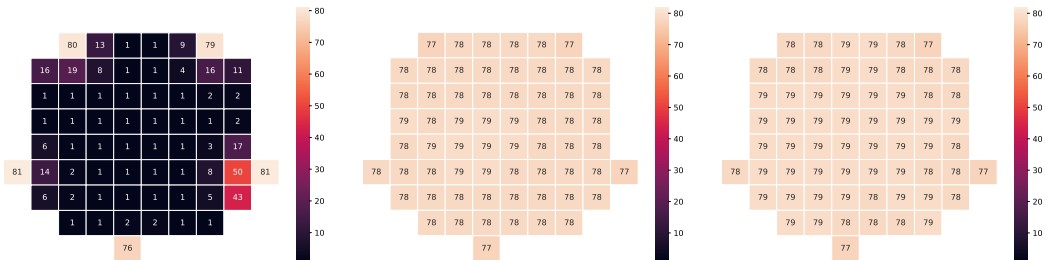

(a) **Accuracy heatmap of the Baseline.** The interpolated models' performance is equal to random ones.

(b) **Accuracy heatmap of WASH.** The accuracy is very similar for various interpolations.

(c) **Accuracy heatmap of WASH+Opt.** The results are similar to WASH.

Figure 6: **Accuracy heatmap for different weight interpolations**, for models trained separately, with WASH or WASH+Opt.

Table 4: **Test accuracies of WASH with variants of the shuffling probability per depth.** Trained with a population of 5 models on CIFAR-100 with a ResNet-18. The results show that permuting the first layers is more important than the later layers. Still, a constant probability across layers does not decrease WASH's performance much.

| Proba. at layer | | | Technique | | | | |
|---|---|---|---|---|---|---|---|
| 0 | to | L-1 | Ensemble | Averaged | GreedySoup | Best model | Worst model |
| $10^{-3}$ | ↘ | 0 | 82.22±.38 | 82.15±.22 | 81.94± 0.25 | 80.89±.03 | 78.80±.77 |
| $10^{-3}$ | → | $10^{-3}$ | 82.04±.19 | 81.94±.15 | 81.69±.23 | 80.60±.16 | 78.67±.89 |
| 0 | ↗ | $10^{-3}$ | 81.75±.35 | 81.37±.10 | 81.14±.20 | 80.08±.40 | 78.55±.70 |

## 7 Additional metrics

**Disagreement in function space.** To support our use of the distance to consensus as an accurate metric of diversity in our paper, we also report a more established metric, the model prediction disagreement, as proposed by (12). This value corresponds to the fraction of examples in the validation set where two models disagree on the prediction. In Fig. 7, we report the disagreement for models trained on the four methods considered in this work: the Baseline without communication, PAPA, WASH, and WASH+Opt. We observe the same ranking in the methods as in the distance to consensus: the Baseline models have the highest disagreement, followed by our methods, and PAPA has the lowest. This confirms that WASH produces more diverse models than PAPA. Note that the Baseline has the highest disagreement, but the models cannot be successfully averaged.

**Expected Calibration Error.** In Tab. 5, we report the ECE for all four methods at optimal temperature, showing that WASH provides better-calibrated models than PAPA. We also report ECE values for varying temperatures in Fig. 8.

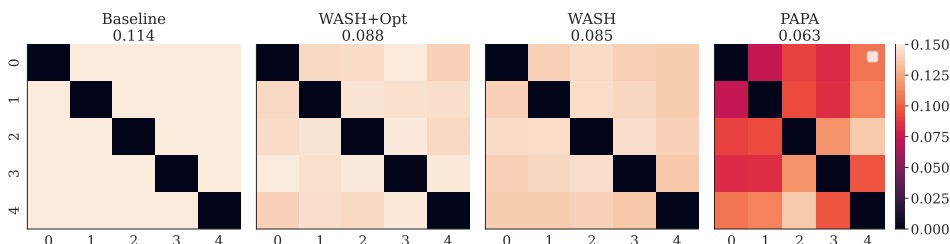

Figure 7: **Disagreement in function space**, for 5 ResNets trained on CIFAR-100 on heterogeneous data. The mean disagreement value for models with different indices is reported on top of the heatmaps. WASH has a higher disagreement between the model predictions (and thus better diversity) than PAPA.

| Method | Indv. | Ens. | Avg. |
|---|---|---|---|
| Baseline | 0.377 | 0.368 | 0.180 |
| WASH | 0.374 | 0.372 | 0.376 |
| WASH+Opt | 0.374 | 0.373 | 0.375 |
| PAPA | 0.376 | 0.376 | 0.378 |

Table 5: **Expected Calibration Error (ECE) for all four methods**, for 5 ResNets trained on CIFAR-100 on heterogeneous data. We report the ECE for the individual models (Indv., averaged for the 5 models), the Ensemble model (Ens.) and the Averaged (Avg.) one. The ECE is the one obtained for the optimal temperature. Our method has a lower ECE than WASH in all cases, showing that it is better calibrated. The very low ECE for the Averaged baseline is due to the fact that the model is close to random.

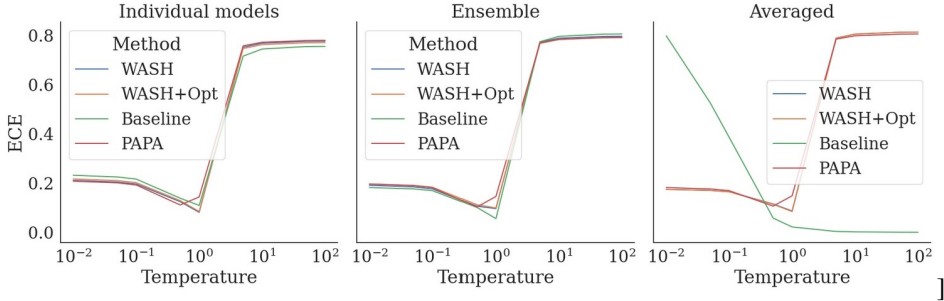

Figure 8: **ECE values for varying temperatures**, for 5 ResNets trained on CIFAR-100 on heterogeneous data. We report the average ECE for the individual models, or for the Ensemble or the Averaged model.

**Communication speed depending on volume** To get a better idea of the potential speed-ups that an effective implementation of WASH could provide, we report in Figure 9 the computation and communication speed of different models with varying factors. We report the average time of a training loop for different batch sizes on ImageNet for a ConvNext tiny or large and a ViT B 16 or L 32. We report the average communication speed of the all-reduce operation of a tensor the size of the model parameters, varying its size when only a fraction $p$ of the parameters are communicated. We consider A100 GPUs connected by an Intel Omni-PAth network (OPA) network (and therefore with a very high connection speed). Even in this case, if we consider several nodes (for 16 or 32 GPUs with 2 or 4 nodes in these cases), we observe that the time to communicate an entire model becomes non-negligible and can be longer than a training loop (here we have not considered simple speed-ups such as the torch.compile code or using mixed precision, for example). However, by communicating only a fraction of the parameters at each step, the communication time would be negligible compared to the computation time even in the worst case.

# 8 Additional results

**ImageNet32x32.** In Tab. 7, we report the accuracy for the dataset ImageNet32x32, showing that a lower PAPA EMA frequency compared to what was reported in their article and code ($T = 10$), results in a better Averaged performance, reproducing their results but still resulting in worse results than WASH. We also find similar results by decreasing the value of the EMA $\alpha$. This confirms that the low performance of our replication of PAPA on ImageNet mainly stems from its hyperparameters, and reinforces our conclusion on the improvements provided by WASH.

We also report in Tab. 6 the accuracy of PAPA on ImageNet for varying EMA frequencies. We find that models finish in the same loss basin when EMA steps are applied every 1 or 2 steps, contrary to what was reported. Thus, to obtain models that can be weight averaged, the actual communication volume improvement provided by WASH would be 5 or 10 times higher than the one reported in Tab. 1.

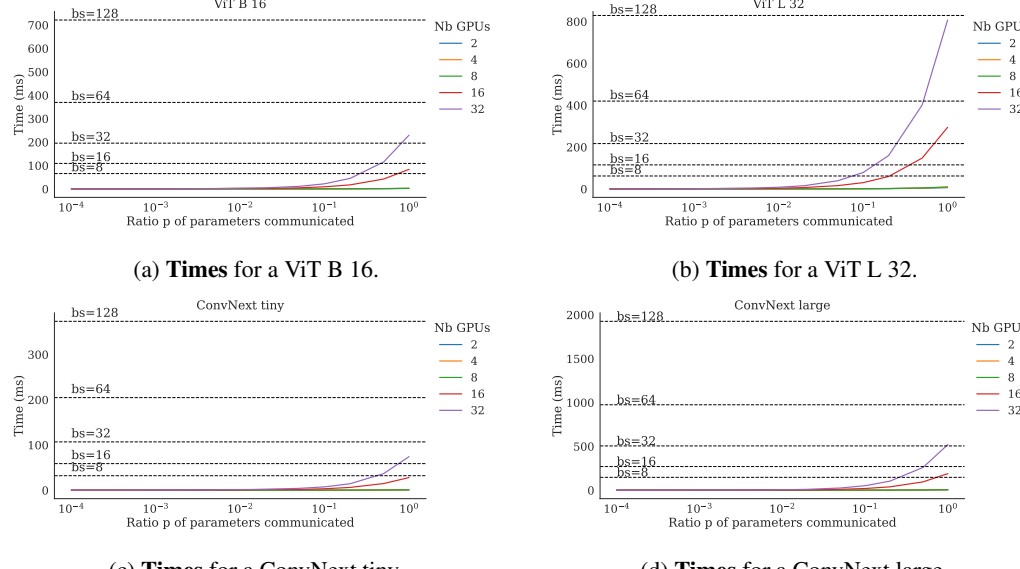

(a) **Times** for a ViT B 16.

(b) **Times** for a ViT L 32.

(c) **Times** for a ConvNext tiny.

(d) **Times** for a ConvNext large.

Figure 9: **Communication and computation speeds**, in average, for a ViT or ConvNext model. We report the mean training loop time for varying batch sizes on ImageNet. We also report the mean communication time of a tensor of the size of the model's parameters, resized by a ratio $p$. In particular for the larger models, when training on separate nodes (16 or 32 GPUs), the communication time can be as long as the computation time. Dividing the communication volume allows a similar divide in the communication time, hiding back the communication.

| $T$ | 1 | 2 | 3 | 4 | 5 | 6 | 7 | 8 | 9 | 10 |
|---|---|---|---|---|---|---|---|---|---|---|
| **Ensemble** | 75.0±.1 | 75.0 | 74.4 | 74.8 | 75.2 | 75.4 | 75.6 | 75.9 | 75.9 | 76.0 |
| **Averaged** | 74.9±.1 | 74.9 | 2.8 | 0.5 | 0.2 | 0.2 | 0.1 | 0.2 | 0.1 | 0.1 |

Table 6: **Performance on ImageNet of PAPA for varying EMA frequencies $T$.** We report the results for 3 runs for $T = 1$. We find that EMA steps every 2 training steps at least are necessary for models to be in the same loss basin.

**REPAIR.** In Tab. 9, we show that the addition of REPAIR further reduces the gap between WASH and the Baseline ensemble accuracy, demonstrating that further post-training techniques (like self-distillation or Stochastic Weight Averaging (SWA) (5; 19)) could further improve our method.

| Method | Baseline | WASH | WASH+Opt | **PAPA** ($T = 10$) | $T = 9$ | $T = 5$ |
|---|---|---|---|---|---|---|
| **Ensemble** | **74.95**±0.95 | 67.55±0.22 | 67.95±0.66 | 61.01±0.31 | 61.34±0.19 | 61.52±0.45 |
| **Averaged** | 0.1±0.0 | 67.80±0.16 | **68.22**±**0.71** | 1.98±1.54 | 35.43±13.67 | 61.05±0.32 |

Table 7: **Performance on ImageNet32**, for all methods on 3 ResNet-50 trained on heterogeneous data. $p = 0.05$ like on ImageNet. We find similar results for PAPA. However, reducing the EMA frequency $T$ allows for a better Averaged accuracy, while still being heavily under WASH's performance.

| N | Method | WASH | WASH+Opt | PAPA |
|---|---|---|---|---|
| 3 | **Averaged** | **81.90±0.19** | **82.08±0.09** | **81.53±0.13** |
|   | **GreedySoup** | 81.73±0.27 | 81.42±0.55 | 80.91±0.74 |
| 5 | **Averaged** | **81.97±0.28** | **82.17±0.15** | **82.01±0.34** |
|   | **GreedySoup** | 81.83±0.26 | 81.49±0.91 | 81.67±1.03 |
| 10 | **Averaged** | **82.31±0.38** | **82.18±0.22** | **82.15±0.14** |
|    | **GreedySoup** | 81.92±0.53 | 81.99±0.17 | 81.92±0.22 |

Table 8: **GreedySoup performances** for WASH and its variant and PAPA, for Resnets-18 trained on CIFAR-100 in the heterogeneous case. GreedySoup is the same method as Diwa. In the case here where averaging all models provides the best results, GreedySoup may only keep a subpar subset of weights to average (generally only one).

| Method | Ens. | Avg. | +REPAIR |
|---|---|---|---|
| Baseline | **83.8** | 0.01 | 0.01 |
| WASH | 82.7 | 82.5 | 82.7 |
| WASH+Opt | 82.4 | 82.5 | **82.8** |
| PAPA | 81.8 | 81.8 | 82.3 |

Table 9: **Effect of REPAIR on the four methods**, for 5 ResNets trained on CIFAR-100 on heterogeneous data. We note that REPAIR has no effect on the Baseline models. Our method's performance can be improved even closer to the baseline Ensemble by using post-training methods like REPAIR.

