# OpenReview forum: "WASH: Train your Ensemble with Communication-Efficient Weight Shuffling, then Average"
_ICLR.cc/2025/Conference — Submitted to ICLR 2025_

### Official Review · Reviewer_9gTH · 2024-10-25

**Soundness:** 3
**Presentation:** 4
**Contribution:** 1
**Rating:** 3
**Confidence:** 3

**Summary:**

This paper proposes WASH, a weight averaging method for obtaining a weight-averaged model with minimal communication cost during the training procedure. The key idea is to randomly shuffle a small fraction of the parameters. Experimental results demonstrate that the proposed method achieves state-of-the-art classification results with a reduced communication burden.

**Strengths:**

* I believe this paper addresses an impactful problem; model averaging with minimal communication cost is promising for democratizing ensemble methods in real-world tasks.
* The proposed method is convincing and intuitive. Moreover, the paper provides empirical analyses that reveal how the method works, which helps the reader understand it.
* The paper is generally well-written and easy to follow.

**Weaknesses:**

I do not find any major flaws in this paper, except in the experimental section. I would like to focus on the experiments on the ImageNet dataset, as I believe one of the main advantages of these model averaging methods is that they can be easily applied to large data regimes and large-scale models compared with probabilistic methods such as Bayesian neural networks.

* The accuracy of the proposed method is significantly lower than the PAPA baseline by 1.23 pp (75.80% for the PAPA ensemble vs. 74.39% for the WASH ensemble method).
* The paper only provides results with small-sized models (e.g., ResNet-50 and smaller). Including results with modern neural networks such as Vision Transformers and ConvNeXt would strengthen the paper.
* I wouldn't say the communication improvement is significant, as WASH still uses between 1/2 and 1/4 of the communication volume compared with PAPA. Providing scenarios or real-world numbers would improve the paper. For example, when we train ResNet-50 on ImageNet, how much communication bandwidth and training time can we save? Do we have to shuffle the models every step to achieve high accuracies?
* The performances of the proposed method, PAPA, and the deep ensemble baseline on ImageNet-1K are all lower than I expected. I anticipated the performance would be higher than 76% even without utilizing ensemble methods [1]. If we utilize modern data augmentations, the accuracy should be about 80% [2].
* It is a minor point, but I consider the performance difference between WASH and WASH+Opt as a weakness. Providing the results for one method and placing the others in the appendix may improve clarity.
* Also, solely providing an image classification task might not be sufficient to demonstrate the effectiveness of the proposed method.

I enjoyed reading about the method and the analyses. However, the performance improvements in terms of accuracy as well as communication cost are not groundbreaking when we consider the high standards of the conference.



[1] https://pytorch.org/vision/0.18/models/generated/torchvision.models.resnet50.html
[2] Wightman, Ross, Hugo Touvron, and Hervé Jégou. "Resnet strikes back: An improved training procedure in timm." *arXiv preprint arXiv:2110.00476* (2021).


---

**Aftre rebuttal.** I believe the proposed method has the potential for impact, but I am not fully convinced by the experimental results or their reliability as presented in the current manuscript.

**Questions:**

Please refer to the weaknesses section.

---

> ### Author Response · Authors · 2024-11-21
>
> We thank the reviewer for their review.
>
> **W1**: It is important to note that our goal in this work is not to achieve better Ensemble accuracy, but to achieve better performance for the Averaged model, or at least comparable performance for a much smaller communication volume.
>
> As discussed with reviewer QJmV, we have provided novel ImageNet experiments in this rebuttal, and that to obtain models in the same loss basins with PAPA, we find that it is necessary to perform EMA steps every 1 or 2 training steps. In this case, we note in particular that the Ensemble accuracy is around 74.7%, very close to that obtained in WASH. Note again that our ImageNet experiments on PAPA are reproduced from the initial codebase of PAPA, and we were not able to retrieve their results compared to CIFAR-100.
>
> **W3**: As mentioned above, we found that the EMA step frequency with PAPA had to be less than 2 to be in the same loss basin. Therefore, the improvement in communication volume is actually between 10 and 40 times that of PAPA. This volume improvement is not negligible and can lead to significant improvements in training speed.
>
> The actual training time improvement will depend on a myriad of factors (computation time, number of workers, communication hardware, and node sizes...), but we refer to [A] for a study showing the impact of communication time on training time for different factors (note that the communication bandwidth in localSGD is directly analogous to DART and PAPA, and thus also to WASH). In these practical settings, the communication time can easily overshadow the computation time. Thus, reducing the communication volume (and thus the communication bandwidth) by a factor of at least $10$ in these cases will result in a significant increase in training speed. Note that we are proposing to shuffle the weights at every step, but only a very small percentage of them. Thus, the communication bandwidth required by WASH is very small, since each worker only needs to send/receive a fraction $p$ of its weights.
>
> We also added a novel experiment in Figure 9, measuring the training computation speed of different models compared to the communication speed for varying numbers of GPUs and communication volumes. We observe that communication, as reported by [A], can slow down training, even in clusters with high-speed connections. However, by limiting the communication volume, the communication time is similarly reduced.
>
> **W4**: We agree with the reviewer that our ImageNet baseline is a few points shorter than it could be. However, our aim here was to compare ourselves to a reproducible baseline. Thus we started from the official code base of PAPA and ran their code with their reported hyperparameters for the ImageNet experiments as it is our main point of comparison.
>
> **W5**: Our aim in presenting the two WASH variants is to show several points: that WASH can be adapted in several ways without affecting performance much; and that the effect of the optimizer's weight shuffling is mainly between the heterogeneous and homogeneous settings. Presenting the results of the two variants side by side makes it easier to observe these points. We will try to improve the clarity of the presentation of WASH+Opt in the final version.
>
> **W2/6**: We agree that more extensive experiments on other datasets and networks could further confirm the effectiveness of our method. Our goal in this work was to present the simplest form of our method and to demonstrate its effectiveness despite its simplicity, as well as to provide a deeper understanding of its mechanisms. Nevertheless, we argue that providing experiments on image classification, in particular on ImageNet, has been sufficient to confirm the results of many other methods in the field, such as in [B-G].
>
> [A] Trade-offs of Local SGD at Scale: An Empirical Study, Ortiz et al., 2021.
>
> [B] Averaging weights leads to wider optima and better generalization, Izmailov et al., 2019.
>
> [C] SWAD : Domain Generalization by Seeking Flat Minima, Cha et al., 2021.
>
> [D] PopulAtion Parameter Averaging (PAPA), Jolicoeur-Martineau et al., 2023.
>
> [E] Learning Neural Network Subspaces, Wortzman et al., 2021.
>
> [F] Loss Surface Simplexes for Mode Connecting Volumes and Fast Ensembling, Benton et al., 2021.
>
> [G] REPAIR: REnormalizing Permuted Activations for Interpolation Repair, Jordan et al., 2023.

---

> ### Comment · Reviewer_9gTH · 2024-11-25
> **RE: Official Comment by Authors**
>
> After reading the author's response, I have additional questions. I believe there may have been some miscommunications. To clarify, I acknowledge that the paper's objective is not to achieve better ensemble accuracy but to enhance the performance of the averaged model. One of my main concerns is the reliability of the experiments.
>
> I think it would be beneficial to paraphrase my questions as follows:
>
> * Why are all the results (including separate training and PAPA) on ImageNet training lower than the standalone training result of 76%? I believe more than a 1pp decrease in accuracy on ImageNet is significant. I initially thought this degradation stemmed from training without data augmentation, but if I understand correctly from the response, that was a typo and simple data augmentations are used. Is that right?
> * If there are some experimental configuration differences, would it be possible to achieve higher accuracy than the standalone training result by using the standalone training configuration or using the modern configurations? Considering that the goal of model ensembling is to improve accuracy and robustness, the inability to use a specific configuration suggests that the performance is limited, which implies that the method's impact is constrained.

---

> > ### Comment · Reviewer_3ghR · 2024-11-26
> >
> > > I believe more than a 1pp decrease in accuracy on ImageNet is significant.
> >
> > I strongly agree that a decrease of 3%p in accuracy on ImageNet is significant. While the authors describe the ResNet-50 results as showing a "small discrepancy" compared to well-established numbers, I respectfully disagree.
> >
> > Furthermore, considering that the performance improvement of the proposed WASH over the baseline is within 3%p, referring to this as "a difference of a few %" seems questionable. Also, since the observed improvements occur at 73% accuracy, it is uncertain whether similar gains would be seen under the standard scenario (76% accuracy with standard augmentation).
> >
> > The 76% accuracy for ResNet-50 on ImageNet with standard augmentation has been consistently reproduced since He et al. (2016). Reporting only 73% accuracy under these conditions may suggest a potential issue in the experimental setup. Given the fully empirical nature of this study, this discrepancy is debatable.

---

> > > ### Author Response · Authors · 2024-12-02
> > >
> > > Dear reviewers,
> > >
> > > We thank the reviewers for highlighting this issue. We have identified the source of our highlighted accuracy discrepancy, which is related to the porting of the PAPA implementation to a distributed setting: the batch size used in our implementation was 8192 instead of 256, with a learning rate that was therefore also not adapted to this framework. The remaining time of the rebuttal did not allow us to run experiments with the small batch size used in PAPA. We will provide updated performance in the small batch size case in the final version, but we agree with the reviewers that the performance given in the ImageNet case is therefore not sufficient to show the efficiency of WASH in this case. Nevertheless, we argue that our experiments in the CIFAR10 and CIFAR100 frameworks are sufficient to demonstrate our claims, as are our further ablations and experiments.

---

### Official Review · Reviewer_3ghR · 2024-11-03

**Soundness:** 2
**Presentation:** 2
**Contribution:** 2
**Rating:** 3
**Confidence:** 3

**Summary:**

This work proposes the WASH algorithm for weight-averaging multiple models trained in parallel under a from-scratch training scenario (i.e., each model starts with different random initializations). Naively trained models from these different initializations converge to different basins on the loss landscape, and thus they cannot be weight-averaged. The WASH algorithm addresses this issue by employing a simple strategy that mixes subset of weights between models during training, enabling them to converge into the same basin and thus be effectively weight-averaged.

**Strengths:**

- The concept behind the proposed WASH algorithm is straightforward and intuitive. Moreover, it incorporates various engineering considerations to improve practical performance, such as adjusting shuffling probability based on depth.
- Sections 1 and 2 provide a comprehensive overview of existing work on ensembling and weight-averaging in the fields of machine learning and deep learning.
- Figure 1 and Algorithm 1 present the proposed WASH algorithm in a clear and easily understandable way.

**Weaknesses:**

- It seems the authors argue that the core of the WASH algorithm is communication efficiency. However, the current manuscript only discusses the theoretical communication volume presented in Table 1. According to main tables (Tables 2 and 3), the WASH algorithm does not show a significant performance advantage over the existing PAPA algorithm. To demonstrate the utility of the WASH algorithm, it would be helpful to present actual communication costs alongside the performance metrics. For instance, providing the wall-clock time required for training could enhance this analysis.

- The main tables (Tables 2 and 3) presents the results of weight-averaging for independently trained models, including UniformSoup (labeled as "Averaged") and GreedySoup. These results are redundant, as none of the combinations achieve effective weight-averaging in such cases. Furthermore, GreedySoup merely reflects the performance of the single model with the best validation metric. Notably, the GreedySoup algorithm relies on the assumption that fine-tuned weights, derived from the same pre-trained model, lie within a single basin suitable for weight-averaging—an assumption valid in transfer learning contexts but not applicable to the from-scratch training scenario in Table 3.

- The ultimate goal of the WASH algorithm is not only to facilitate weight-averaging of multiple models trained in parallel through parameter shuffling but also to ensure that the resulting weight-averaged model achieves superior performance. Therefore, it is crucial to compare the WASH algorithm with other high-performance single-model approaches, particularly those based on weight-averaging (such as single-trajectory strategies for from-scratch training, including Stochastic Weight Averaging by Izmailov et al. (2018) and the Lookaround optimizer by Zhang et al. (2023)). However, the only baseline currently presented is PAPA, along with the fundamental baselines of a single model representing a performance lower bound and an ensemble model (commonly referred to as deep ensembles) representing a performance upper bound.

- Recent trends have significantly shifted from from-scratch training to transfer learning approaches that involve fine-tuning pre-trained models. For instance, the model soup paper referenced in this study focused on fine-tuning both supervised and CLIP pre-trained vision models for ImageNet in 2022, yielding extensive experimental findings. Since this work is exclusively centered on a from-scratch training scenario, the lack of fine-tuning experiments with contemporary pre-trained models makes the results less compelling in 2024.

**Questions:**

Here are my questions and further concerns:

- I understand that the homogeneous setup is described as not using data augmentation, as noted in line 315, which states "homogeneous setting with no data augmentations." However, the performance of the single model presented in Table 3, specifically the performance of GreedySoup—given that it achieves the best validation metric for a single model in a from-scratch training scenario—appears excessively high. Could it be that basic data augmentation techniques, such as random cropping with random horizontal flipping, were applied instead of no augmentation at all?

- According to Table 2, the accuracy of the single ResNet50 model on ImageNet is approximately 74%. Given that there are no modifications to the ResNet50 architecture and that training data augmentation was applied, this result seems quite unusual. For example, the V1 ResNet50 checkpoint from torchvision achieves around 76% when trained for 90 epochs with a batch size of 256 and without a complex training recipe. Since this paper employs advanced augmentation techniques—such as mixup, label smoothing, cutmix, and random erasing—we would expect the performance to exceed 76%. In fact, the PAPA paper (Jolicoeur-Martineau et al., 2024) demonstrates that a single ResNet50 model achieves an accuracy of 76.8% on ImageNet under these conditions. Therefore, there are concerns about the appropriateness of the experiments conducted, in addition to the earlier mentioned issues regarding the no-data-augmentation in the homogeneous setup.

- Is there a specific reason why the ImageNet results are absent from the homogeneous setup? Currently, the ImageNet results are only presented in Table 2 and not in Table 3.

- Figures 2 and 3 do not need such a large allocation of space, as they lack small elements that require close examination. As they stand, they occupy excessive space.

- The conclusion section is missing a section number; it should be labeled as "5. Conclusion" rather than just "Conclusion."

- How statistically significant is the claim in Appendix B.3 that WASH outperforms PAPA in terms of ECE for the CIFAR-100 scenario? Additionally, since Table 4 mentions the "optimal temperature," it appears that the ECE was computed after applying temperature scaling. How does the ECE compare without temperature scaling? Also, it might be worth considering NLL as an additional metric for evaluating predictive uncertainty.

- The results in Appendix C.5 suggest that using UniformSoup (i.e., averaging) yields better performance than GreedySoup for the multiple soup ingredients obtained with WASH. Do you have any educated guesses as to why this might be the case? Why could selecting only the weight-averaged ingredients based on validation metrics result in worse performance?

---

> ### Author Response · Authors · 2024-11-21
>
> We thank the reviewer for their thorough review.
>
> **W1**: First, note that we have performed additional experiments on PAPA's ImageNet (for $N=3$) by varying the frequency of EMA steps. We find that the models converge to the same loss basin only when EMA steps are performed every 1 or 2 steps. Thus, the effective communication volume improvement of WASH over PAPA is actually between 10x and 40x (depending on the number of steps and whether or not WASH+Opt is used).
>
> In this paper, we report the theoretical communication volume improvement as an objective way to measure the improvement of our method. In comparison, the effective training speed improvement depends on many factors, such as computational speed (model/batch size, hardware...) and communication speed (number of nodes, inter/intra node connections...). For these reasons, we decided to focus on a value agnostic to the training framework.
>
> However, we agree with the reviewer that it remains important to measure this type of improvement in practice. We first refer the reviewer to another paper that measures the impact of communication on training throughput in the case of localSGD (which is functionally equivalent to, for example, DART/PAPA) [A]. In their framework, they find that the time taken by communication is much higher than the time taken by computation, even in cases with only 16 workers. Note also that the recorded communication time increases linearly with the number of workers, and decreases similarly with fewer averaging steps (and thus a smaller communication volume).
>
> We also added a novel experiment in Figure 9, measuring the training computation speed of different models compared to the communication speed for varying number of GPUs and communication volumes. We observe that communication, as reported by [A], can slow down training, even in clusters with high-speed connections. However, by limiting the communication volume, the communication time is similarly reduced, hiding once again the communication time.
>
> **W2**: As pointed out by the reviewer, we report the Averaged and GreedySoup accuracies for the Baseline models, trained separately. Our aim here is twofold. Firstly, reporting the Averaged accuracy aims to clarify that in all cases considered here, even with $p=3$, homogeneous models starting from the same initialisation, the converged models do not end up in the same loss basin. This is not a novelty, of course, but it serves to make this important point clear. Adding the GreedySoup accuracy gives some additional clarifying information for the rest of the Table: it shows that standard merging approaches provide worse accuracies than methods like PAPA or WASH, which explicitly train models to be in the same loss basin. By retaining only the accuracy of a single model, it is also analogous to reporting the top accuracy of a single model trained separately, an important baseline for comparing the Averaged models trained by PAPA and WASH.
> We argue that this presentation is not a weakness of our paper, but a way of effectively presenting: that the separately trained models are in different loss basins and therefore unmergeable, with individual model accuracies that are outperformed by the Averaged models.
>
> **W3**: Indeed, in this work, we have not directly compared ourselves with methods such as SWA or the Lookaround optimizer, as these methods are orthogonal and fully complementary to our approach. These techniques can be used to improve the generalisation of each model during the training phase by finding flatter minima in their optimisation path. Note that SWA in particular was compared to PAPA in Appendix A.11 of their paper. They showed that SWA provides a similar improvement to PAPA, and that combining the two approaches leads to even further performance gains. We expect WASH to behave similarly, as our mechanism works similarly to the EMA steps used in PAPA. Our goal in this work was not to find the most effective combination of techniques (such as REPAIR, which we find improves performance, or other optimizers such as SWA or Lookaround) to achieve state-of-the-art performance, but to introduce a novel distributed training algorithm that greatly decreases communication costs  and compare ourselves with similar approaches such as PAPA or DART.

---

> > ### Author Response · Authors · 2024-11-21
> >
> > **W4**: Our aim in this paper is to present a completely novel training technique and to provide an in-depth explanation of it, with multiple ablations, metrics and additional experiments. We have also shown our improvements over another state-of-the-art method, PAPA, with experiments similar to those used in their paper. We agree with the reviewers that more extensive experiments, with more finely tuned hyperparameters and schedulers, could be performed in a separate paper to definitively confirm the effectiveness of our approach on multiple benchmarks, but this was not the goal of this paper. Note also that our approach is particularly suitable for pre-training models, a task that is more computationally intensive than fine-tuning, and where a high communication overhead is even more undesirable.
> > Although fine-tuning has become more popular in recent years, pre-training approaches remain a crucial part of the training process and continue to merit specific study.
> >
> > **Q1**: This is indeed an error, and we thank the reviewer for pointing it out. We do apply simple random cropping and flipping to the data in all datasets, which adds some slight heterogeneity (and improves performance). This oversight will be corrected in the final version.
> >
> > **Q2**: For a fair comparison with PAPA's experiments, we used the same code base as they provided to reproduce their examples (note that DART did not provide code for their experiments for comparison). Our ImageNet framework is derived from theirs, and we did not modify any hyperparameter except the number of EMA steps (see our new experiments, where we find accuracies close to 75% for PAPA, less than 2% away from the reported one).
> > Nevertheless, we agree that we could increase the accuracy of our ImageNet experiments by using a more modern training framework, such as that of [C].  Despite this small accuracy gap with the state of the art, we argue that our experiments still show that WASH performs similarly to PAPA, for a fraction of the communication volume.
> >
> > **Q3**:  There was no particular reason, and we agree with the reviewer that they may be of interest. We have added new experiments (for $N=3$) for ImageNet in the homogeneous case, with EMA steps at each training step in the case of PAPA. In this case we find that WASH again gives better results than PAPA. Results for $N=5$ will be added for the final version.
> >
> > **Q4/5**: Thank you, these will be changed in the final version (see the modified submission).
> >
> > **Q6**: Indeed, here, we have chosen an optimal temperature value for the calibration error, as is standard practice (see e.g. Section 4.c "Procedure" of [B]). In Figure 8 of the Appendix, we report the ECE values for different temperatures to give a better understanding of the effect of temperature on the ECE calculation. The comparison using the full range of temperatures is similar and we again find in this case that WASH is better calibrated than PAPA, although the improvement here is not statistically significant.
> >
> > **Q7**: This is indeed a counterintuitive result. To gain a better understanding, the heat maps in Figure 6 may be helpful. Consider, for example, that all the models are close to the edges of the loss basin, with the Averaged model close to the center of the basin. Following the GreedySoup  algorithm, only one model at a time is added to the soup. If the two models selected are not on "opposite sides" of the basin, then note that the average of these two models, even with high individual validation accuracy, may be worse, even though the average of all models is better. This is the same reason why averaging over a larger number of models allows us to achieve better performance. The early stopping of GreedySoup, which is effective when models may not be in the same loss basin, becomes a disadvantage in our case where it would be better to use all models. Note that the authors of PAPA have also confirmed this result.
> >
> > [A] Trade-offs of Local SGD at Scale: An Empirical Study, Ortiz et al., 2021
> >
> > [B] DICE: Diversity in Deep Ensembles via Conditional Redundancy Adversarial Estimation, Ramé et al., 2021.
> >
> > [C] Resnet strikes back: An improved training procedure in timm, Wightman et al., 2021.

---

> ### Comment · Reviewer_3ghR · 2024-11-23
>
> > Nevertheless, we agree that we could increase the accuracy of our ImageNet experiments by using a more modern training framework, such as that of [C]. Despite this small accuracy gap with the state of the art, we argue that our experiments still show that WASH performs similarly to PAPA, for a fraction of the communication volume.
>
> The concerns regarding the results for ResNet50 on ImageNet are not about achieving high performance but also about the impact on the reliability of the experimental results presented in the paper. The top-1 accuracy of the baseline reported in the PAPA paper is 76.8%, a reasonable improvement of around 1%p over the expected 76% range with standard augmentation and 90 epochs of training, probably thanks to the use of advanced data augmentation techniques. However, the baseline performance reported in this paper is around 74% with the same advanced data augmentation (and around 73% in a homogeneous setting with standard augmentation), suggesting a significant issue with the experimental setup, which calls into question the validity of any experimental comparisons in this context.

---

> > ### Comment · Reviewer_9gTH · 2024-11-24
> > **RE: Official Comment by Reviewer 3ghR**
> >
> > I believe this point aligns with my main concerns, and I strongly agree with this comment. I would greatly appreciate it if the authors could address this concern in detail.

---

> > > ### Author Response · Authors · 2024-11-25
> > >
> > > We agree with the reviewers that there is a difference (of a few %) in accuracy between our experiments and those reported by PAPA in their article on ImageNet.
> > >
> > > Importantly, we must point out that our experiments for PAPA are based on the codebase provided by the PAPA paper, in order to make a fair comparison. We are not aware of the reasons for the small discrepancy between their reported results and our reproductions of their results, using their codebase. Note that we also had to modify the EMA frequency to obtain models in the same loss basin with PAPA on ImageNet. Still, this small decrease in accuracy affects the results of all methods equally, and thus none of the methods are favored by it. Note also that our aim with WASH is not to claim that our approach necessarily offers better performance than other approaches such as DART/PAPA-all or PAPA in all cases. Our goal is to provide a completely novel distributed training approach, with extensive experiments and ablations to gain a better understanding of it; and to show that this approach offers comparable or in most cases better performance than approaches like PAPA for a drastic reduction in communication volume (and thus in training time).
> > >
> > > We believe that in this work, we offered extensive experiments and ablations to demonstrate the novelty and effectiveness of our method, and that this small reduction on one dataset, kept for the sake of a fair comparison, does not affect our conclusions and contributions.

---

### Official Review · Reviewer_RDjK · 2024-11-04

**Soundness:** 3
**Presentation:** 3
**Contribution:** 3
**Rating:** 6
**Confidence:** 4

**Summary:**

The paper proposes a novel method, wash, for training deep ensembles with two key selling points:
* The individual models remain in the same weight basin, meaning that robust predictions can be made by averaging the weights of the individual members and executing a single forward pass.
* Wash requires a fraction of the communication cost incurred in each training step compared to similar approaches.

WASH works by shuffling a small percentage of the weights between the ensemble members in each training iteration. The result is that the members become robust to this augmentation and by the end of training, the weights remain near each other (in the same basin).

Experimental results show that WASH matches and often exceeds the performance of PAPA at a much lower communication cost. The models are evaluated on image classification benchmarks: Cifar10, Cifar100 and ImageNet, across three architectures: VGG16, ResNet18 and ResNet50.

**Strengths:**

The paper proposes a very interesting and novel approach for keeping the ensemble members near each other in weight space, while keeping the communication cost low.

The method clearly works: There is a clear improvement in communication cost without a significant negative impact on performance. The image classification benchmarks show improvement over PAPA for multiple architectures.

The paper is very well presented: The paper also dives into the intuition of why WASH works. It explains the dynamics of the networks through a 2d toy example. It also showcases that WASH has a better diversity by examining the distances of the individual models from the consensus throughout training.

The experiments are thorough: The paper ablates three design choices: the layerwise adaptation of the shuffle probability, the magnitude of the shuffle probability and the idea of using a warmup period instead of enabling shuffling from iteration 1.

**Weaknesses:**

I was disappointed that the experiments did not detail the impact of the reduced communication cost. The work is primarily motivated by the reduction in communication cost, but we don't see experiments showcasing the overall compute reduction during training or the improvement in training time. It is impressive that the communication cost is reduced by a factor of 1/2-1/200, but it is unclear whether this results in compute savings or faster training.

**Questions:**

* Addressing the weakness mentioned above.

* The choice of a linear schedule for the shuffle probability is rather arbitrary. The ablation in the appendix examines a constant vs increasing vs decaying schedule and we see that the decaying schedule gives an improvement. Are there experiments with other schedules? I wonder if p could be tuned per layer based on the variance of the weights across the ensemble members.

* Why do the imagenet experiments require so much larger p than the cifar experiments? Is there an intuitive reason for this? Should we expect that as models and data scale, p will get closer to 1?

* What is the distance metric used in Figure 2?

---

> ### Author Response · Authors · 2024-11-21
>
> We thank the reviewer for their review.
>
> **W1/Q1**:
> Our aim in this submission was to provide a novel method that improves on previous ones for a lower communication volume, rather than precisely measure an improvement in training speed. Such an improvement will depend on many factors, such as the size of the model and the batch, the hardware speed, the communication speed (which also depends on the number of nodes, the type of inter- and intra-node connections, etc.), and more. For these reasons, in this work, we preferred to focus on the performance of the method (with further ablations and experiments) and on the communication volume, a simple value to compare methods, that is agnostic to other variables such as the type of hardware.
>
> Nevertheless, we fully agree with the reviewer that it remains important to measure such improvements in practice. We refer the reviewer first to another paper that measures the impact of communication on training throughput in the case of localSGD (which is equivalent to PAPA, for example) [A]. Note that in their case the communication time is higher than the computation time, even for cases with 16 workers. Note also that the communication time increases linearly with the number of workers, and decreases significantly as fewer averaging steps are needed (and thus a smaller communication volume).
>
> We also added a novel experiment in Figure 9, measuring the training computation speed of different models compared to the communication speed for varying number of GPUs and communication volumes. We observe that communication, as reported by [A], can slow down training, even in clusters with high-speed connections. However, by limiting the communication volume, the communication time is similarly reduced, hiding once again the communication time.
>
> Finally, as discussed with reviewer QJmV, note that we added further experiments on ImageNet for PAPA and found comparable accuracies to WASH only with EMA steps every 1 or 2 steps, resulting in an even higher communication volume required.
>
> **Q2**: Other schedules have been considered. We tried using different fixed values in different parts of the networks, for example dividing it into quarters. Instead of decreasing to $0$, we also tried decreasing to higher values, with worse results and higher communication volumes. In general, these experiments all converged on the same trend: more permutations early in the networks and fewer for the later layers. Our chosen schedule was thus chosen to be simple and applicable to a wide variety of networks, without the need for other hyperparameters.
>
> The idea suggested by the reviewer is quite interesting and seems to be a way to effectively choose more precise schedules for different architectures, as well as noting which components require more shuffling. It would also be consistent with our intuition. Note, however, that knowledge of parameter variance is not applicable during training, as it would require communication between workers to obtain this information. We emphasize that contrary to PAPA, our method WASH never performs an averaging operation requiring an all-reduce communication and only permutes parameters; thus accessing the variance is not possible without adding a significant communication overhead. It remains a very interesting idea to explore and we thank the reviewer.
>
> **Q3**: First, note that larger models do not directly require a higher $p$ value: using ResNets-50 with the same $p$ as smaller models also leads to the same loss basin when training on CIFAR-100 (see Table 6).
>
> The need for a higher $p$ value for ImageNet compared to CIFAR-100 seems analogous to the need for more frequent EMA steps in PAPA, as we have found (see our changes and discussion with reviewer QJmV). In both cases, more actions are needed to keep the models in the same loss basin. There are several possible reasons for this: sharper local minima, more local minima with larger data sets, etc. Nevertheless, it suggests that more complex datasets require more work to keep the models in the same loss basin.
>
> However, we do not believe that it will be necessary to reach values of $p$ close to $1$. The most communication required for a model trained with approaches such as PAPA or DART is to require an EMA step at each step, which is already the case for ImageNet here. Even in this case, we achieve the same accuracy for a small value of $p$. In a case where $p$ becomes too high, a simple workaround could be to combine weight shuffling steps with periodic and infrequent EMA or averaging steps. This approach would allow us to maintain a small communication volume while keeping the models in the same loss basin.
>
> **Q4**: The distance metric reported here is a simple L2 distance between parameters. The consensus is computed and we compute the average distance between the weights of each model and the weights of the consensus.
>
> [A] Trade-offs of Local SGD at Scale: An Empirical Study, Ortiz et al., 2021

---

> > ### Comment · Reviewer_RDjK · 2024-11-25
> > **Thank you for the reply. Looking for more detail on the compute savings.**
> >
> > Dear Authors,
> >
> > Thank you for the detailed reply.
> >
> > Q1: I examined the plots in the revision, and they showcase that the communication cost indeed decreases with lower values of p as expected. While I understand that there are many factors to consider (such as the specific hardware at hand), I am looking get an intuition on *how much compute savings can a practitioner expect by implementing the proposed approach*.
> >
> > My question asked for "experiments showcasing the overall compute reduction during training". Could you provide a figure or table showing the overall training compute cost (in gpu hours) comparing PAPA vs WASH for the models shown in Tables 2 and 3?
> >
> > In the revision I see
> > * "We report the average communication speed of the all-reduce operation of a tensor the size of the model parameters, varying its size when only a fraction p of the parameters are communicated." --> This is Figure 9.
> > * "We report the average time of a training loop for different batch sizes on ImageNet for a ConvNext tiny or large and a ViT B 16 or L 32." --> Where can I find this in the revised manuscript?
> >
> > Q2: Thank you for the reply. I think it would be nice to also mention these details in the manuscript.
> >
> > Q3: Thank you for the reply. This answers my question.
> >
> > Q4: Thank you for the reply. Please add a clarification on how the distance is computed in the Figure caption.

---

> > > ### Author Response · Authors · 2024-12-02
> > >
> > > We apologise for the lack of clarity in our Figure 9. The average time of a training loop is shown as the dashed horizontal black lines in Figure 9 for different batch sizes; in contrast, the coloured lines represent the communication speed.
> > >
> > > More generally, the computational savings that a practitioner can hope to achieve are highly dependent on the framework under consideration. In cases with a very limited number of models (such as $3$), where models are trained on the same node in a cluster, this improvement will be negligible. In cases where several nodes are considered (for $16$ or $32$ GPUs in our figure), i.e. for nodes with less than $4$ GPUs, which will already be the case when training at least $5$ models, the communication time can vary between, in the best case, about $10%$ of the computation time; but it can very easily reach cases where it reaches $100%$ and more of the computation time (i.e. both times are comparable). In the lower cases, the improvement provided by WASH will collapse this value to negligible, reducing the training time by about $10%$. But in the highest cases, if WASH can divide the computational volume (and thus its time) by even $10$, the training speed can be doubled or more.

---

### Official Review · Reviewer_QJmV · 2024-11-12

**Soundness:** 3
**Presentation:** 4
**Contribution:** 3
**Rating:** 6
**Confidence:** 4

**Summary:**

This work proposed a shuffling-based ensemble training algorithm that enables immediate weight averaging with good generalization ability without the need of further finetuning. The proposed method is motivated by the fact that a previous method called PAPA will gradually reduce the distance of the models to their consensus, thus hidering their diversity. In contrast, the proposed shuffling operation preserves the consensus distance and thus potentially improves model diversity and performance. Moreover, it also requires less communication and thus improves communication efficiency.

**Strengths:**

1. The paper is clearly written and the related work section is thorough and very informative.
2. Besides the main results in Tab 2 & 3, the authors also provided other informative empirical results to show why the proposed method is effective and superior to PAPA, like the observation of consensus distances and the 2D example.

**Weaknesses:**

1. The authors mainly compared the proposed method with PAPA. I understand that maybe this work is a close follow-up on PAPA and thus PAPA is the main competitor. But I checked the results in the PAPA paper and found that PAPA itself can be inferior to DART. So I think it would be better to include DART in the comparisons or at least discuss why not including it.

2. Advantages over PAPA are marginal most of times. The gaps are only significant with larger models and larger datasets.

3. While there are reproduction issues with PAPA on ImageNet, I think it would be better to also include those results in Table 3 and faithfully explain how the results went wrong, considering that there is still spare space for the main text.

**Questions:**

I think Tab 1 is only a conceptual illustration of the communication efficiency of WASH. In real world, do the authors think that the randomness in WASH will make it harder for really efficient hardware implementation?

---

> ### Author Response · Authors · 2024-11-21
>
> We thank the reviewer for their review.
>
> **W1**: As the reviewer points out, the results reported in the DART paper seem to outperform the EMA method of PAPA in some cases. However, as the PAPA authors point out in Section 4.5.2, DART does not provide an implementation and the hyperparameters used are unclear. Furthermore, they find that in their own re-implementation, the accuracy of DART is lower than reported. Notably, it performs worse than PAPA-all (which is fundamentally the same method). Since the authors of PAPA report that their PAPA-all method generally performs worse than PAPA, we have chosen to report only PAPA over DART and PAPA-all in order to keep only the best (reproducible) performing method.
>
> **W2**: Indeed, the improvement gap appears mainly in the most difficult cases, on larger datasets and with a higher number of models. These are precisely the cases we are interested in, and in particular where the communication volume can become a bottleneck for faster training. Note also that the remaining gap with the Ensemble may be due to the fact that all models are in the same loss basin, which directly limits their diversity (see line 353).
>
> **W3**: We agree with the reviewer and have extended our experiments and discussion regarding the reimplementation in Appendix C.4, reporting results on ImageNet for different EMA frequencies in Table 6.  We found that training on ImageNet required an EMA step every 1 or 2 steps, rather than every 10 steps as reported. We found accuracies very close to those of WASH, for a communication volume 40 times higher.
>
> **Q1**: This is an interesting point, and we thank the reviewer for discussing it. First, note that the randomness is predetermined by a random seed that is known to all workers in advance, so no communications are required to know which weights are being shuffled and between whom. Then note also that the only computations that need to be done for the shuffling step (other than the communications) are: randomly choosing which parameters to permute (which can be done efficiently with a random Bernoulli mask, for example), and choosing permutations for each one. This can be a bit slower, but remember that only a very small proportion of the parameters are shuffled at each step. To be absolutely sure that no computational overhead is caused by this entire step, it is even possible to do it in parallel with the training computations, as it requires a very small amount of memory (only about one layer in memory at a time for efficient computations, and each worker only needs to store the few shuffled parameter indexes and their associated previous/next workers in the permutation).

---

### Author Response · Authors · 2024-11-21
**Modifications and new experiments.**

We thank all reviewers for their thorough comments and suggestions. We have added several new experiments and made changes to address some concerns.

**Imagenet PAPA accuracy and homogeneous framework** First, we provided new ImageNet results for PAPA with different EMA step frequencies. We found that EMA steps are needed at least every 2 training steps for the models to reach the same loss basin, with accuracies similar to WASH.
We also added results for ImageNet in the homogeneous case (with an EMA frequency of $1$ step for PAPA) and found that WASH also outperforms PAPA in this case.

**Impact of communication volume on training speed** Our aim in this work was to provide an objective improvement in communication volume, as reporting the resulting improvement in training speed will depend on many factors. Nevertheless, we agree with the reviewers that it was necessary to provide some measurements to show the possible improvement resulting from the smaller communication volume, and we provide some experiments, measuring computation and communication times for different models, batch sizes, number of GPUs and nodes, and by varying the ratio of parameters computed. We find that reducing the communication volume can mask the impact of communication on training speed when training on multiple nodes.

We also added expected calibration error values for different temperatures, not just the optimal ones. Finally, we also modified the description of the homogeneous settings, reduced the size of Figures 2 and 3 and added the number of the Conclusion section.

---

### Comment · Area_Chair_rGH4 · 2024-11-24
**Reminder - Public Discussion Phase Ending Soon**

Dear PC memebers,

Thank you for your valuable comments during the review period, which raised many interesting and insightful questions. Now the discussion period is coming to a close, please take a moment to review the authors’ responses if you haven’t done so already. Even if you decide not to update your evaluation, kindly confirm that you have reviewed the responses and that they do not change your assessment.

Timeline: As a reminder, the review timeline is as follows:

November 26: Last day for reviewers to ask questions to authors.
November 27: Last day for authors to respond to reviewers.
November 28 - December 10: Reviewer and area chair discussion phase.
December 20: Meta reviews and initial decisions are due.


Thank you for your time and effort!

Best regards,
AC

---

### Meta-Review · Area_Chair_rGH4 · 2024-12-18

**Metareview:**

This paper proposes an intuitive method to enhance diversity while maintaining the same basin by shuffling a small percentage of weights for ensemble learning. The primary advantage highlighted is improved communication efficiency. Although the overall scores show some divergence, the reviewers' concerns are consistent: while the basic idea is generally accepted, doubts remain about the experiments. These include unconvincing results on ImageNet, lack of comparisons with stronger baselines, and unclear advantages in computational time. The discussions on this paper have been thorough and ultimately converge toward rejection. We hope the reviewers’ comments and discussions will help the authors further refine and improve their method.

**Additional Comments On Reviewer Discussion:**

The discussions were quite active. While the overall ratings differ, the underlying concerns are largely consistent: all reviewers agreed that the experiments were insufficient. In general, the authors failed to address these concerns convincingly. Reviewer 9gTH even lowered their score after a critical question regarding the ImageNet results was not adequately addressed. This concern was also echoed by Reviewer RDjK, who initially gave a score of 6 but finally suggested rejection following the internal discussion.

---

### Decision · Program_Chairs · 2025-01-22

Reject